# Household Air Pollution Is Associated with Chronic Cough but Not Hemoptysis after Completion of Pulmonary Tuberculosis Treatment in Adults, Rural Eastern Democratic Republic of Congo

**DOI:** 10.3390/ijerph15112563

**Published:** 2018-11-15

**Authors:** Patrick D.M.C. Katoto, Aime Murhula, Tony Kayembe-Kitenge, Herve Lawin, Bertin C. Bisimwa, Jean Paul Cirhambiza, Eric Musafiri, Freddy Birembano, Zacharie Kashongwe, Bruce Kirenga, Sayoki Mfinanga, Kevin Mortimer, Patrick De Boever, Tim S. Nawrot, Jean B. Nachega, Benoit Nemery

**Affiliations:** 1Centre for Environment and Health, Department of Public Health and Primary Care, KU Leuven, 300 Leuven, Belgium; tonykayemb@gmail.com (T.K.-K.); tim.nawrot@uhasselt.be (T.S.N.); ben.nemery@kuleuven.be (B.N.); 2Department of Internal Medicine and Prof Lurhuma Biomedical Research Laboratory, Mycobacterium Unit, Faculty of Medicine, Catholic University of Bukavu, 02BP, Bukavu, Congo; draime2006@yahoo.fr (A.M.); bcasinga@gmail.com (B.C.B.); 3Unit of Toxicology and Environment, University hospital of Lubumbashi, School of Public Health Faculty of Medicine, Lubumbashi 1825BP, Congo; 4Unit of Teaching and Research in Occupational and Environmental Health, Faculty of Health Sciences, University of Abomey-Calavi (UAC), Cotonou 03BP0490, Benin; hervelawin@gmail.com; 5Département de Biologie médicale, Institut Supérieur des Techniques Médicales (ISTM) Bukavu, BP 3036, Bukavu, Congo; 6National TB Program, Provincial and national Anti-Leprosy and TB Coordination, BP. 1899, Bukavu, Dem. Congo; jeanpaulchira@gmail.com (J.P.C.); emulume@gmail.com (E.M.); freddybirembano@yahoo.fr (F.B.); zkashongwe@yahoo.fr (Z.K.); 7Department of Pulmonary Medicine and Lung Institute, Makerere University, PB 7072, Kampala, Uganda; brucekirenga@yahoo.co.uk; 8National Institute for Medical Research Muhimbili Medical Research Centre, PB 65001, Dar es Salaam, Tanzania; gsmfinanga@yahoo.com; 9Liverpool School of Tropical Medicine, Liverpool L3 5QA, UK; Kevin.Mortimer@lstmed.ac.uk; 10Centre for Environmental Sciences, Hasselt University, Agoralaan, building D, 3590, Diepenbeek, Belgium; patrick.deboever@vito.be; 11Health Unit, Flemish Institute for Technological Research (VITO), Vlasmeer7, 2400 Mol, Belgium; 12Department of Medicine and Center for Infectious Diseases, Faculty of Medicine and Health Sciences, Stellenbosch University, 8000, Francie Van Zijl Drive, PB 241, Cape Town, South Africa; jnachega@sun.ac.za; 13Departments of Epidemiology and International Health, Johns Hopkins Bloomberg School of Public Health, Baltimore, 21205, MD, USA; 14Departments of Epidemiology, Infectious Diseases and Microbiology, University of Pittsburgh Graduate School of Public Health, Pittsburgh, 15210, PA, USA

**Keywords:** biomass fuel, kerosene, respiratory symptoms, post-pulmonary tuberculosis, South Kivu

## Abstract

Little is known about the respiratory health damage related to household air pollution (HAP) in survivors of pulmonary tuberculosis (PTB). In a population-based cross-sectional study, we determined the prevalence and associated predictors of chronic cough and hemoptysis in 441 randomly selected PTB survivors living in 13 remote health zones with high TB burden in the South Kivu province of the Democratic Republic of Congo (DRC). Trained community and health-care workers administered a validated questionnaire. In a multivariate logistic regression, chronic cough was independently associated with HAP (adjusted odds ratios (aOR) 2.10, 95% CI: 1.10–4.00) and PTB treatment >6 months (aOR 3.80, 95% CI: 1.62–8.96). Among women, chronic cough was associated with cooking ≥3 h daily (aOR 2.74, 95% CI: 1.25–6.07) and with HAP (aOR 3.93, 95% CI: 1.15–13.43). Independent predictors of hemoptysis were PTB retreatment (aOR 3.04, 95% CI: 1.04–5.09) and ignorance of treatment outcome (aOR 2.24, 95% CI: 1.09–4.58) but not HAP (aOR 1.86, 95% CI: 0.61–5.62). Exposure to HAP proved a major risk factor for chronic cough in PTB survivors, especially in women. This factor is amenable to intervention.

## 1. Introduction

With an estimated 10.4 million new cases and over 1.7 million deaths globally in 2016, tuberculosis (TB) remains of great public health importance [1]. In the Democratic Republic of Congo (DRC), the 2015 targets for TB control have not been met, i.e., neither the incidence nor the prevalence or mortality of TB were reduced by 50% compared to the situation in 1990 [2]. Consequently, in this country with a high TB burden (World Health Organization, global tuberculosis report 2018) [3], the number of pulmonary TB (PTB) survivors continues to increase. After completion of TB treatment, complications such as obstructive airways diseases have been documented since the days of the sanatoria treatment [4]. It is unclear why such a residual morbidity has remained neglected in many countries. National TB programs, even from high-burden TB countries, do not integrate long-term follow-up data of PTB survivors, nor do they provide information about complications in post-PTB life [1,2,5,6].

Although effective cures exist for the treatment of active TB disease, PTB survivors may still face various long-term complications from the architectural compromise of lung parenchyma (residual cavitation, scarring, or fibrosis, with mainly restrictive functional impairment) or airways (bronchiectasis, tracheobronchial stenosis, broncholithiasis, with obstructive impairment) due to colonization and infection (mycetoma or fungal ball) [5]. It is conceivable that exposure to household air pollution (HAP) increases the morbidity caused by the structural and functional impairment resulting from the previous mycobacterial infection. Exposure to HAP caused by the use of unclean domestic energy (biomass fuel, kerosene) has been associated with several lung illnesses [6]. Among HAP, wood is the most commonly used in Africa and especially in the Democratic Republic of Congo (DRC), which harbors the second largest forest in the world. In the DRC, with a rural electrification rate of 0.4%, the population relies on biomass fuel [7] for cooking and even for heating during the night time (mountain regions) and rainy seasons. Inadequately burnt wood generates a complex mixture of carbon-based particles, inorganic particles, irritant gases, and carcinogens, as in tobacco smoke [6,8].

Tuberculosis is associated with about 1.2 quality-adjusted life years lost, with 80% related to disability after TB cure. Of the disability-adjusted life years (DALYs) lost due to TB, 77% are linked to pulmonary impairment after successful treatment [9]. On the other hand, the comparative risk assessment for the 2010 Global Burden of Disease lists HAP as the third highest risk factor for lost DALYs in the world. Deaths and DALYs due to HAP are very unequally distributed; the DRC is among the top-10 worst-affected countries, where HAP has been estimated to be responsible for more than 1.5 million yearly deaths [10]. The burden of TB and that of HAP are related and inextricably linked to poverty. Hence, one can anticipate that HAP adds substantial morbidity for PTB survivors and costs for health systems. In resource-limited countries such as the DRC, with high TB burden and widespread use of biomass for domestic energy, we hypothesized that among PTB survivors, exposure to HAP would be associated with chronic complications, as assessed by chronic cough and or hemoptysis. Our study aimed at providing quantitative data on the impact of HAP in PTB survivors and bring the issue of exposure to this modifiable risk factor under the attention of the community at large.

## 2. Materials and Methods

### 2.1. Design and Settings

We conducted a community-based cross-sectional study in the South Kivu province, in the eastern DRC, from 24 May 2017 to 17 March 2018. The South Kivu province has a surface area of 65,128 km² (source: Geographic Institute, DRC). Its main city, Bukavu, is situated 2000 km from the capital city, Kinshasa. The average population density is 101 inhabitants/km^2^ with a maximum of 250 inhabitants/km^2^ in lakeside areas and a minimum of 10 inhabitants/km^2^ in forest and savanna areas. The province has 34 health zones (HZs), with 113 clinics integrating both diagnosis and treatment of TB and 405 clinics offering TB treatment alone. In this study, we included 13 rural HZs based on their 2016 TB incidence and/or accessibility (Figure 1).

### 2.2. Participants

As shown in the flow diagram (Figure 2), PTB survivors older than 15 years were selected from the TB clinics 2016 registers of the selected rural HZs by simple random sampling (25% of the total) using Stata 14 (Mersenne Twister, a default random-number generator). Only those declared cured or having completed treatment were included. We excluded subjects transferred to another hospital, those having failed treatment, those lost to follow-up, and those deceased. We also excluded participants treated for extra-pulmonary TB and those undergoing PTB treatment during the study period. Criteria for time elapsed between cure and inclusion was not considered since previous studies showed no particular duration for the onset or peaking of respiratory illnesses among PTB survivors [9,11,12]. Well-trained community workers known as the “*ambassadeurs de la lutte contre la TB*” screened for potential eligible participants in the community, explained the purpose of the study, presented the consent form, and assisted junior doctors in performing the interview. The “*ambassadeurs de la lutte contre la TB*” are members of a community-based non-governmental organization engaged in the advocacy for and fight against TB. They are PTB survivors themselves or family members of PTB survivors. They regularly work closely with the national TB program and are involved in community-based activities of TB active case-finding and directly observed treatment (DOT) strategies [13]. The survey was conducted using an interviewer-administered questionnaire in the native language of the PTB survivors using commonly understood terms for all variables. A total of three languages were used in the survey. No effort was made to clinically test for the disease or its complications. 

### 2.3. Instrument

We adapted the Inter-Multidisciplinary Program to Address Lung Health and Tuberculosis in Africa (IMPALA) questionnaire to assess sources of energy for cooking, heating and lighting, and symptoms [14].

### 2.4. Predictor Variables

Exposure to HAP from biomass fuel smoke and kerosene was ascertained indirectly by type of energy used for domestic needs (cooking or heating or lighting). Of note, kerosene is not used for cooking in this area but only for lighting and this has decreased significantly due to the introduction of light-emitting diodes (LEDs) [8] in the past decade. Other variables included personal data (age, sex, marital status), living conditions (house type, kitchen type, livestock kept indoors, poverty, tobacco smoke), and health data (comorbidity and history of TB).

### 2.5. Operational Definition of Variables

Marital status: married vs unmarried (=single, widowed, separated). Kitchen location (with regards to the main house): outdoors = outside and separated from the main house, outside but adjoining the house; indoors = inside the house but separate from other rooms, inside the house but not separated from other rooms; both = indoors and outdoors (concurrently or alternately). Kitchen ventilation: yes = presence of chimney or door/window that can open to the outside; no = no chimney and no door/window that can directly open to the outside. Time spent in the kitchen per day: ≤3 h/day and >3 h. Domestic Energy: HAP (or unclean energy) = any wood, charcoal, straw, kerosene used for cooking, heating or lighting (in the kitchen or in the house); clean energy = electricity or liquefied petroleum gas used for cooking or heating or lighting, without parallel use of any of the fuels defined above as unclean. Tobacco smoke: yes = current smoker, ex-smoker or any regular smokers in the same house or at work (these categories being grouped because active smoking was infrequent, and even non-existent among women); no = none of the above. Poverty: the index was constructed using the following variables: thatched roof, walls in mud or wood, absence of electricity or presence of electricity but time spent with electricity per day ≤2.5 h, house with ≤2 rooms, crowding (≥4 inhabitants in the house), livestock (including cattle, goats, rabbits, guinea pigs, poultry) kept indoors, and no mattress on the bed. To score for socioeconomic status (SES), we did not use conventional scores because some low-cost devices are widely available in this setting. For example, one might find a radio or a mobile phone worth around United States Dollar (USD)5–10 that does not reflect the true SES of the study participants. Vermin/pests in the house or on the bed: yes = rats/mice, bed ticks/roaches; no = not present. Room light: bedroom lighting by the sun during daytime = yes/no. Comorbidity: diabetes mellitus = yes/no; hypertension = yes/no; Human immunodeficiency virus infection (HIV) = yes/no. Awareness of PTB treatment outcomes: cured or not cured (treatment completion or unknown decision). TB retreatment: yes = if the participant completed two or more TB treatment regimens. Hospitalization during PTB treatment: yes = if participant was hospitalized at least once during the TB treatment period. TB treatment period: six or more than six months.

### 2.6. Outcome Variables

Respiratory symptoms were defined based on responses from the study participant’s interview. Symptoms included chronic cough and hemoptysis. To characterize chronic cough, PTB survivors were asked to report any cough of ≥4 weeks duration since completing TB treatment. To report for hemoptysis, PTB survivors were asked to report if they had seen any blood in the sputum since completing their TB treatment.

### 2.7. Statistical Analysis

We calculated the prevalence of chronic cough and hemoptysis by type of household energy use, by gender and by other included covariates. We summarized normally distributed continuous variables by their mean and standard deviation (±SD), and used a two-sample independent *t*-test for comparing mean values. Categorical variables were summarized by counts and percentages, and we used Pearson’s chi-square (chi2) to compare differences between groups. We used univariate and multivariate logistic regression modelling to identify predictors independently associated with odds of chronic cough or hemoptysis. A bivariate logistic regression model was first fitted, and the variables significant at *p*-value < 0.2 in the bivariate analysis were used in the final multivariate logistic regression, in addition to a priori selected variables clinically or epidemiologically relevant. Crude and adjusted odds ratios (aORs) and associated 95% confidence intervals (CI) were calculated to summarize the strength of association between baseline characteristics and chronic cough or hemoptysis. Stratified analyses were conducted to explore potential gender differences in the associations between exposure to HAP and chronic cough or hemoptysis. This sub-analysis was performed due to the known gender inequality in the role of cooking. Independent variables in the logistic regression models were considered to be statistically significant predictors of respiratory symptoms if *p* < 0.05. All *p*-values were two-sided. Analyses were performed using Stata 14 (Stata Corp, College Station, TX, USA).

### 2.8. Ethics Clearance

Ethical approval was obtained from the Institutional Review Board of the Catholic University of Bukavu in the frame of the “Environmental Exposure and Risk of Respiratory Illnesses in Kivu (EERRIK Project)” (UCB/CIE/NC/01/2018). A letter of support was obtained from the South Kivu Provincial Health Department. Informed consent was obtained from all study participants. 

## 3. Results

### 3.1. Covariate Distribution Among PTB survivors by Gender

Table 1 summarizes the baseline characteristics of the 441 PTB survivors included in the study.

Of the 441 study participants, 298 (67.6%) were males, their mean age was 45 years (SD 15) with almost half of participants aged between 35 and 55 years. About 31% of females were unmarried, compared to 19% of males. Fifty percent of kitchens were located indoors or both indoors and outdoors, with 19% of these indoor kitchens having no chimney or ventilation. As expected, a larger proportion of women spent more than 3 hours cooking than men (43% vs 32%). Only 16% of PTB survivors were not exposed to HAP from cooking, heating, or lighting, without gender difference. Nearly all participants (>98%) were poor, based on the type of dwelling, crowding, etc. Overall, 18.6% of participants had been hospitalized for TB, the proportion being higher among women (25.9%) than among men (15.1%).

### 3.2. Reported Prevalence of Chronic Cough and Hemoptysis in PTB Survivors by Gender on Selected Covariates

Table 2 summarizes self-reported frequencies of chronic cough and hemoptysis among PTB survivors in the whole group and separately for men and women in a bivariate analysis.

Chronic cough was self-reported by 56% of participants, without difference in prevalence between men (57%) and women (53%). Among women with chronic cough, 57% (43/76) declared spending more than three hours daily in the kitchen, which is higher (chi2 = 12.85, *p* < 0.001) than among women without chronic cough (18/67, 27%). Hemoptysis was self-reported by 8% of participants, without difference in prevalence between men and women. For hemoptysis, the only significant difference was found with regard to knowledge of outcome after completing PTB treatment among men, where men with hemoptysis were more frequently ignorant (13/23, 57%) than those without hemoptysis (95/275, 35%) (chi2 = 4.43, *p* = 0.035).

### 3.3. Effects of HAP (Smoke from Biomass Fuel and Kerosene) and other Predictors on Chronic Cough or Hemoptysis among Former PTB Patients, by Gender

Table 3 and Table 4 summarize the association between HAP and other covariates and chronic cough and hemoptysis.

In the multivariate analysis, chronic cough was significantly associated with exposure to HAP (aOR 2.10, 95% CI 1.10–4.00) and with PTB treatment for more than 6 months (aOR 3.80, 95% CI 1.62–8.96) in the whole group of PTB survivors. Among women, the odds of reporting chronic cough were increased by approximately 4-fold (aOR 3.93, 95% CI 1.15–13.43) when exposed to HAP and by approximately 3-fold (aOR 2.74, 95% CI 1.25–6.07) when spending more than three hours daily for cooking. Among men, PTB treatment of more than six months increased the odds of chronic cough by approximately 4-fold (aOR 4.45, 95% CI 1.64–12.09) but HAP did not show significant difference between groups (aOR 1.61 [0.74–3.50]).

Independent predictors of hemoptysis among PTB survivors were PTB retreatment (aOR 3.04, 95% CI 1.04–5.089) and ignorance of treatment outcomes after completion of PTB treatment (aOR 2.24, 95% CI 1.09–4.58). Women retreated for PTB (aOR 8.71, 95% CI 1.00–77.17) and men who ignored the clinical decision after completion of their PTB treatment (aOR 2.44, 95% CI 1.01–5.88) were at greater risk of suffering from hemoptysis than did new cases and those who knew their clinical outcomes, respectively.

## 4. Discussion 

The main original finding of our cross-sectional study into possible environmental risk factors of chronic respiratory illnesses (chronic cough and hemoptysis) among PTB survivors is that exposure to HAP was significantly associated with chronic cough, especially among women. No such relation was observed for hemoptysis, a complication that we found to be more likely among those who had not undergone an optimal TB treatment.

Respiratory complications after anti-TB completion are diverse and well documented [5]. To our knowledge, this is the first investigation reporting on the association between exposure to HAP and respiratory symptoms in a population of PTB survivors. Our findings are consistent with previous studies highlighting a high rate of utilization of unclean fuels (biomass fuel and kerosene) in rural settings from low and middle-income countries (LMICs) [8], as well as gender inequality with regard to exposure to HAP (daily time spent in kitchen) [8,15,16] and associated adverse respiratory health effects [8,15,17,18,19]. With two-thirds of our randomly chosen PTB survivors consisting of men, our study population is consistent with other studies reporting a higher proportion of male individuals among PTB survivors [20]. In our population with a very low prevalence of tobacco smoking [21], HAP strongly predicted chronic cough among women. A study pooling data from population-based cohorts from 13 LMIC countries [22] recently suggested that exposure to HAP was the leading population attributable risk factor for chronic obstructive pulmonary disease (COPD), even above that of cigarette smoking. The effect estimate was also high in women for whom 21% of COPD prevalence was due to HAP exposure. Tobacco and biomass smoke are likely to share the same composition as they are both generated from combustion of plants. On the mechanistic view, as does tobacco smoke, biomass smoke also increases the expression of some of the same matrix metalloproteinases. In addition, as for tobacco, our study underscores the effect of exposure-time (time spent in the kitchen for > 3 h per day) in the occurrence of respiratory symptomatology. The vulnerability of women for chronic exposure to indoor biomass and kerosene smoke has been abundantly demonstrated in LMICs [15,16]. In general, hormonal (menopause vs. not) and biological status (higher inspiratory flow in women vs. men) and speculative differences in epithelial response might explain this vulnerability observed in women compared to men. Moreover, specific determinant related to LMICs setting that might explain women’s vulnerability to HAP should be highlighted. First, due to socio-economic structure, women in LMICs are more likely to be nearer the source of HAP than are men. They are responsible for cooking and might spend more than seven hours daily by the fire. Second, those exposed to HAP are at higher risk of presenting systemic inflammation than those using liquid petroleum gas. Finally, yet importantly, women in LMICs are at risk of suffering from anemia (multiple pregnancy) that might increase susceptibility to infection if exposed to HAP [6,8,23].

In the DRC, the burden of PTB is still high in spite of the global effort to reduce PTB [24]. In this study, major factors associated with hemoptysis were related to PTB morbidity. TB can cause chronic impairment of lung function, the severity of which increases with the number of episodes of diseases [12]. This is compatible with the independent association we found between hemoptysis and retreatment. In retreated PTB patients, the shrinking of immune function and the dysbacteriosis resulting from the repetition of using large spectra antibiotics both favor fungal colonization. In this setting, data on relapse with or without acquired drug-resistant TB or re-infection of a new Mycobacterium Tuberculosis strains as partial cause of chronic cough post-PTB are scarce. Preliminary data reported by Bulabula and colleagues comparing Rifampicin- resistant (n = 142) to Rifampicin- susceptible (n = 1366) PTB patients identified by Xpert MTB/RIF have demonstrated a high rate of retreatment after earlier default or failure (21% vs. 3%) and of having a history of ≥3 previous episodes of TB (16% vs. 1%) [25]. In environments in which patients are frequently re-infected such as in this rural area, pulmonary impairment progressively worsens with each TB episode [5]. We also found that men ignoring their clinical TB outcome tended to present hemoptysis compared to women and were more likely at increased risk of presenting hemoptysis compared to their counterpart. Our results are comparable to previous findings showing the high rate of default in men compared to women [26] which might jeopardize their clinical outcome [27] and lead to further complications [28]. Hemoptysis can, therefore, be a proxy of post PTB complication such as aspergillosis/ mycetoma, bronchiectasis etc. In this poor remote setting, we had no diagnostic resources (such as radiology, bronchoscopy, spirometry, or laboratory facilities) to characterize the various possible structural anomalies and functional processes underlying chronic cough or hemoptysis. Consequently, no specific treatment can be offered either. Although the background prevalence of chronic respiratory symptoms in the source population is not known, the fact that more than 50% of our adult PTB survivors reported chronic cough and 8% reported hemoptysis does represent an extra threat for the fragile health system. PTB survivors who continue to have respiratory symptoms are often treated again for so-called “bacteriologically negative TB”, whilst their symptoms may simply be due to or exaggerated by their HAP exposure in addition to the post-TB sequelae. On the other hand, with symptoms remaining or appearing after completing a long TB treatment, many people will rely on traditional healers or religious leaders. The additional morbidity risk added by the HAP among PTB survivors is theoretically modifiable [29]. A study from Malawi [30] recently found a significantly lower level of airway macrophage black carbon in women using a cleaner burning biomass-fueled cookstove compared to those using the traditional open fires for cooking.

Strengths of our study include the fact that (1) it is the first community-based, multicentric study (South-Kivu province-wide) assessing the prevalence and associated predictors of chronic cough and hemoptysis among PTB survivors in a setting of high TB burden; and (2) it reports the discovery of the association between exposure to HAP and risk of respiratory symptoms among PTB survivors. Although informative with novel data, limitations of the present work should also be considered. Firstly, both exposures and outcomes were self-reported. This could introduce risks of recall bias and non-differential misclassification that might influence the accuracy of the results and bias the results towards the null. We focused here on two relevant symptoms (chronic cough and hemoptysis) and not, for instance, on dyspnea, which was also assessed, because this symptom did not prove sufficiently sensitive without grading its severity. Secondly, absence of exposure to HAP was extremely low (comparator), thus possibly biasing the comparator towards better socioeconomic status, even though the majority of our participants were uniformly poor. Nevertheless, bias with regard to HAP exposure was minimized by expressing exposure not only as a qualitative yes/no variable, but by quantifying its intensity as daily time spent cooking. The higher effect found among women compared to men strengthens the plausibility of a causal relationship between exposure to HAP during cooking and chronic cough. Thirdly, the cross-sectional nature of the study precluded assessing temporality. Prospective studies are warranted to validate the present findings. Longitudinal studies should consider generating “life-time exposure” that should contain changes in exposures through new practices and sources of domestic energy, documented by improved exposure quantification and biomarkers of exposure. Exposure quantification in the kitchen as well as urinary biomarkers of exposure to tobacco smoke could specifically help overcoming ascertainment bias that might occur, since some rural kitchens might present a broken wall and as people tend to underreport their tobacco smoking habits. Finally, future studies should consider excluding re-occurrence of TB (relapse or re-infection) as possible cause of chronic cough post-PTB and including pulmonary function testing to improve inference to chronic cough and at least blood spot quantification to confirm hemoptysis.

## 5. Conclusions

In summary, chronic respiratory symptoms are frequent among PTB survivors and HAP proved to be a modifiable independent risk factor. Besides being a feature of active TB, both chronic cough and hemoptysis can also be manifestations of various complications. A programmatic approach to reduce this prevalence should consider four pillars: strengthening the health system, integration of follow-up after PTB treatment in the national TB program, advocacy regarding adverse effects caused by HAP, and consequently efficient interventions to reduce HAP.

## Figures and Tables

**Figure 1 ijerph-15-02563-f001:**
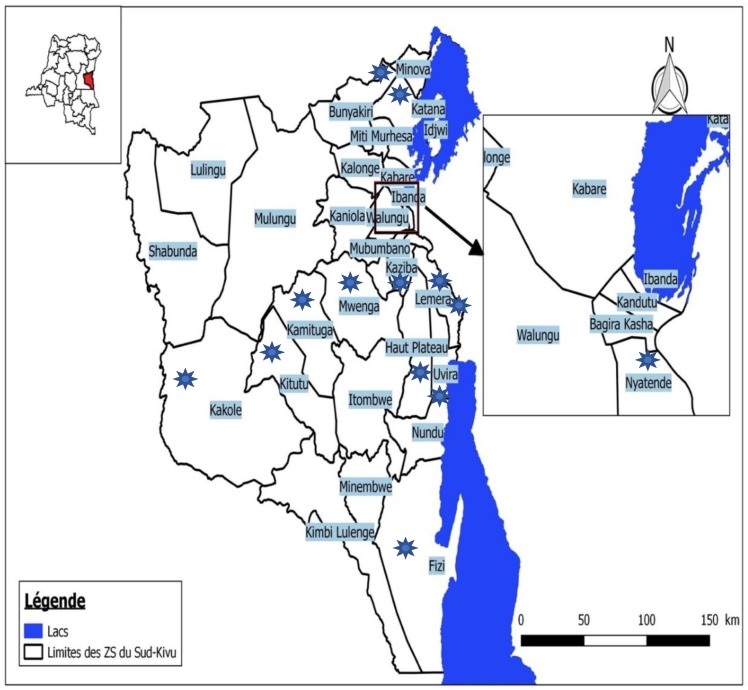
South-Kivu health zones. Stars indicate the health zones (HZs) selected for the studies. Map, courtesy of Dr. Rosine Bigirinama. The cumulative incidence of tuberculosis (TB) in the 13 selected health zones ranged from 16 to 217 per 100,000 inhabitants in 2016.

**Figure 2 ijerph-15-02563-f002:**
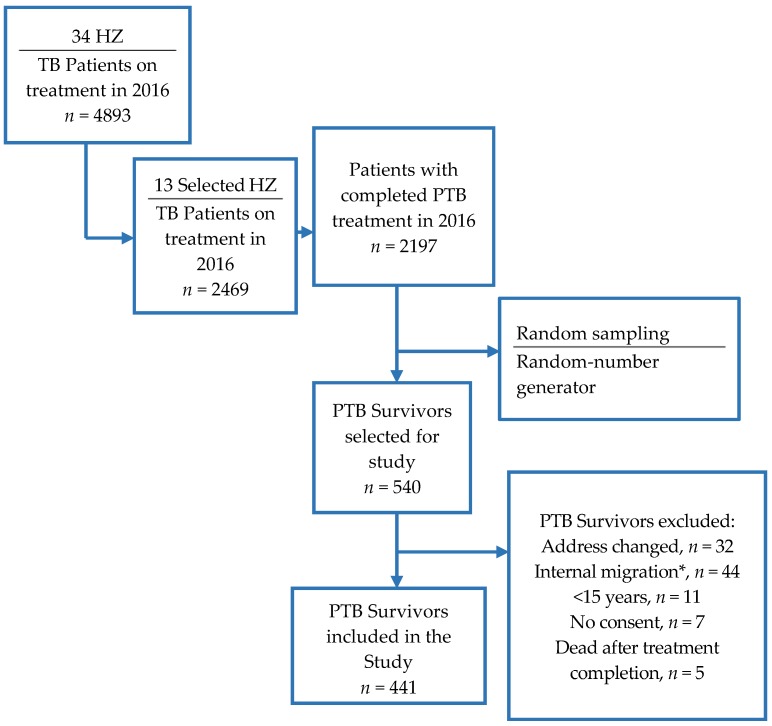
Flow chart. TB: tuberculosis, PTB: pulmonary tuberculosis, HZ: health zone, *: travelling, becoming refugees or displaced for work.

**Table 1 ijerph-15-02563-t001:** Distribution of covariates among 441 survivors of pulmonary TB in rural DRC.

	All *N* = 441	Males *N* = 298 (67.6)	Females *N* = 143 (32.4)	*p*-Value (Males vs. Females)
Age (years), mean (±SD)	44.6 (±14.9)	44.9 (±14.6)	43.9 (±15.6)	0.24
Age group 15–35 years 36–55 years >55 years	121 (27.4) 214 (48.5) 106 (24.0)	71 (23.8) 153 (51.3) 74 (24.8)	50 (35.0) 61 (42.7) 32 (22.4)	0.047
Unmarried	101 (22.9)	57 (19.1)	44 (30.7)	0.006
Kitchen outdoor Indoor + outdoor Indoor	218 (49.4) 52 (11.8) 171 (38.8)	145 (48.7) 34 (11.4) 119 (39.9)	73 (51.1) 18 (12.6) 52 (36.4)	0.76
No kitchen ventilation	82 (18.6)	56 (18.8)	26 (18.2)	0.88
Cooking > 3 h/day	53 (34.7)	92 (30.9)	61 (42.7)	0.015
Unclean energy	368 (83.5)	248 (83.2)	120 (83.9)	0.85
Tobacco smoke exposure	81 (18.4)	54 (18.1)	27 (18.9)	0.85
Poverty	433 (98.2)	292 (98.0)	141 (98.6)	0.65
Vermin/pests in home	406 (92.1)	271 (90.9)	135 (94.4)	0.21
No sunlight in bedroom	193 (43.8)	140 (47.0)	53 (37.1)	0.049
Any comorbidity	113 (25.6)	80 (26.9)	33 (23.1)	0.40
Hospitalization for TB	82 (18.6)	45 (15.1)	37 (25.9)	0.006
TB retreatment	29 (6.6)	21 (7.1)	8 (5.6)	0.57
Ignore treatment outcome	168 (38.1)	108 (36.2)	60 (42.0)	0.25
No DOT	68 (15.4)	47 (15.8)	21 (14.7)	0.77
TB treatment >6 months	49 (11.1)	37 (12.4)	12 (8.4)	0.21

All data, except age, are numbers (with percentages). SD: standard deviation; TB: tuberculosis; DRC: Democratic Republic of Congo; DOT: directly observed treatment.

**Table 2 ijerph-15-02563-t002:** Prevalence of self-reported chronic cough or hemoptysis and of selected covariates in 441 survivors of pulmonary TB in rural DRC.

	Chronic Cough (*N* = 247)			Hemoptysis (*N* = 35)		
	All *N* = 247/441 (56.0)	Males *N* = 171/298 (57.0)	Females *N* = 76/143 (53.1)	All *N* = 35/441 (7.9)	Males *N* = 23/298 (7.7)	Females *N* = 12/143 (8.4)
Age group 15–35 years 36–55 years >55 years	66 (26.7) 122 (49.4) 59 (23.9)	37 (21.6) 89 (52.1) 45 (26.3)	29 (38.2) 33 (43.4) 14 (18.4)	7 (20.0) 18 (51.4) 10 (28.6)	5 (21.7) 10 (43.5) 8 (34.8)	2 (16.7) 8 (66.7) 2 (16.7)
Unmarried	55 (22.3)	33 (19.3)	22 (29.0)	6 (17.1)	3 (13.0)	3 (25.0)
Kitchen outdoor Indoor + outdoor Indoor	127 (51.4) 25 (10.1) 95 (38.5)	88 (51.5) 16 (9.7) 67 (39.2)	39 (51.3) 9 (11.8) 28 (36.8)	17 (48.6) 1 (2.9) 17 (48.6)	12 (52.2) 1 (4.4) 10 (43.5)	5 (41.7) 0 (0.0) 7 (58.3)
No kitchen ventilation	45 (18.2)	32 (18.7)	13 (17.1)	6 (17.1)	4 (17.4)	2 (16.7)
Cooking > 3 h/day	102 (41.3)	59 (34.5)	43 (56.6)	9 (25.7)	3 (13.0)	6 (50.0)
Unclean energy	210 (85.0)	142 (83.0)	68 (89.5)	31 (88.6)	19 (82.6)	12 (100.0)
Tobacco smoke exposure	42 (17.0)	33 (19.3)	9 (11.8)	6 (17.1)	6 (26.1)	0 (0.0)
Poverty	244 (98.8)	169 (98.8)	75 (98.7)	34 (97.1)	22 (95.7)	12 (100.0)
Vermin/pests in home	225 (91.1)	156 (91.2)	69 (90.8)	32 (91.4)	21 (91.3)	11 (91.7)
No sunlight in bedroom	106 (42.9)	79 (46.2)	27 (35.5)	15 (42.9)	10 (43.5)	5 (41.7)
Any comorbidity	74 (30.0)	53 (31.0)	21 (27.6)	10 (28.6)	7 (30.4)	3 (25.0)
Hospitalization for TB	51(20.7)	29 (17.0)	22 (29.0)	6 (17.1)	4 (17.4)	2 (16.7)
TB retreatment	16 (6.5)	13 (7.6)	3 (4.0)	5 (14.3)	3 (13.0)	2 (16.7)
Ignore treatment outcome	89 (36.3)	60 (35.9)	29 (38.2)	19 (54.3)	13 (56.5)	6 (50.0)
No DOT	35 (14.7)	28 (16.4)	7 (9.2)	6 (17.1)	6 (26.1)	0 (0.0)
TB treatment > 6 months	41 (16.6)	31 (18.3)	10 (13.2)	3 (8.6)	2 (8.7)	1 (8.3)

All data are numbers (with percentages); TB: tuberculosis; DRC: Democratic Republic of Congo; DOT: directly observed treatment.

**Table 3 ijerph-15-02563-t003:** Crude and adjusted effects of HAP and other predictors on chronic cough among survivors of pulmonary TB, overall and by gender.

	Chronic Cough					
	Crude OR [95% CI]		Adjusted* OR [95% CI]			
	All	Males	Females	All	Males	Females
Sex	0.84 [0.56–1.26]			0.84 [0.54–1.28]	-	-
Age group 15–35 years 36–55 years >55 years	Ref 1.11 [0.71–1.73] 1.05 [0.62–1.77]	Ref 1.28 [0.73–2.25] 1.43 [0.74–2.76]	Ref 0.85 [0.40–1.81] 0.56 [0.23–1.38]	Ref 1.21 [0.75–1.94] 1.11 [0.64–1.93]	Ref 1.55 [0.85–2.81] 1.71 [0.86–3.40]	Ref 0.76 [0.33–1.73] 0.42 [0.15–1.15
Unmarried	0.92 [0.59–1.44]	1.03 [0.57–1.84]	0.83 [0.41–1.70]	-	-	-
Kitchen Outdoor Indoor + outdoor Indoor	Ref 0.66 [0.36–1.22] 0.90 [0.60–1.34]	Ref 0.58 [0.27–1.22] 0.84 [0.51–1.36]	Ref 0.87 [0.31–2.45] 1.02 [0.50–2.08]	-	-	-
No kitchen ventilation	0.95 [0.58–1.53]	0.99 [0.55–1.78]	0.86 [0.37–2.01]	-	-	-
Cooking > 3 h/day	1.97 [1.31–2.97] **	1.50 [0.90–2.49]	3.55 [1.75–7.18] ***	1.45 [0.93–2.27]	1.06 [0.61–1.85]	2.74 [1.25–6.07] *
Unclean energy	1.29 [0.78–2.14]	0.97 [0.52–1.79]	2.45 [0.97–6.22]	2.10 [1.10–4.00] *	1.61 [0.74–3.50]	3.93 [1.15–13.4] *
Tobacco smoke exposure	0.81 [0.50–1.32]	1.21 [0.66–2.21]	0.37 [0.15–0.88] *	-	-	-
Poverty	2.15 [0.51–9.12]	2.75 [0.50–15.24]	1.14 [0.07–18.53]	-	-	-
Vermin/pests in home	0.73 [0.36–1.50]	1.09 [0.50–2.41]	0.15 [0.02–1.25]	-	-	-
No sunlight in bedroom	0.92 [0.63–1.35]	0.93 [0.59–1.47]	0.87 [0.44–1.71]	-	-	-
Any comorbidity	1.70 [1.09–2.65] *	1.66 [0.98–2.84]	1.75 [0.79–3.90]	1.63 [0.97–2.74]	1.56 [0.83–2.93]	1.74 [0.67–4.53]
Hospitalization for TB	1.37 [0.84–2.24]	1.42 [0.73–2.74	1.41 [0.66–3.02]	1.34 [0.75–2.41]	1.14 [0.53–2.43]	1.98 [0.75–5.24]
TB retreatment	0.96 [0.45–2.06]	1.22 [0.49–3.05]	0.51 [0.12–2.22]	-	-	-
Ignore treatment outcome	0.82 [0.56–1.21]	0.89 [0.56–1.43]	0.72 [0.34–1.40]	-	-	-
No DOT	0.81 [0.48–1.35]	1.11 [0.59–2.10]	0.38 [0.15–1.02]	-	-	-
TB treatment > 6 months	4.63 [2.11–10.1] ***	4.47 [1.80–11.1] **	4.92 [1.04–23.4] *	3.80 [1.62–8.96] **	4.45 [1.64–12.1] **	2.94 [0.50–17.43]

HAP: household air pollution; TB: tuberculosis, DOT: directly observed treatment, Ref = reference; OR: odds ratio, * adjusted for (when not studied) age, sex, source of domestic energy, cooking time, length of TB treatment, hospitalization for TB, any comorbidity; significant ORs in bold; empty cells (-) indicate variables not included in the final model. *p*-value level of significant: * *p* < 0.05; ** *p* < 0.01; *** *p* < 0.001.

**Table 4 ijerph-15-02563-t004:** Crude and adjusted effects of HAP and other predictors on hemoptysis among survivors of pulmonary TB, overall and by gender.

	Hemoptysis					
	Crude OR [95% CI]			Adjusted ^§^ OR [95% CI]		
	All	Male	Female	All	Male	Female
Sex	1.10 [0.53–2.27]	-	-	1.20 [0.57–2.55]	-	-
Age group 15–35 years 36–55 years >55 years	Ref 1.50 [0.61–3.69] 1.70 [0.62–4.63]	Ref 0.92 [0.30–2.81] 1.6 [0.50–5.15]	Ref 3.62 [0.73–17.91] 1.6 [0.21–11.97]	Ref 1.66 [0.66–4.20] 1.70 [0.61–4.74]	Ref 0.99 [0.32–3.09] 1.60 [0.48–5.31]	Ref 5.25 [0.81–34.0] 2.18 [0.22–21.8]
Unmarried	0.68 [0.27–1.68]	0.61 [0.18–2.14]	0.73 [0.19–2.84]	-	-	-
Kitchen Outdoor Indoor + Outdoor Indoor	Ref 0.23 [0.03–1.08] 1.31 [0.65–2.64]	Ref 0.34 [0.04–2.68] 1.02 [0.42–2.44]	Ref NE 2.12 [0.63–7.08]	-	-	-
No kitchen ventilation	0.90 [0.36–2.24]	0.90 [0.30–2.77]	0.89 [0.18–4.34]	-	-	-
Cooking > 3 h/day	0.63 [0.29–1.38]	0.31 [0.09–1.08]	1.38 [0.42–4.51]	0.63 [0.28–1.40]	0.34 [0.10–1.18]	1.28 [0.36–4.59]
Unclean energy	1.59 [0.54–4.64]	0.95 [0.31–2.94]	NE	1.86 [0.61–5.62]	1.07 [0.33–3.41]	NE
Tobacco smoke exposure	0.91 [0.37–2.28]	1.67 [0.63–4.45]	NE	-	-	-
Poverty	0.60 [0.07–4.99]	0.41 [0.05–3.64]	NE	-	-	-
Vermin/pests in home	0.91 [0.27–3.15]	1.05 [0.23–4.74]	0.62 [0.07–5.52]	-	-	-
No sunlight in bedroom	0.97 [0.48–1.93]	0.86 [0.36–2.02]	1.24 [0.37–4.11]	-	-	-
Any comorbidity	1.18 [0.55–2.53]	1.21 [0.48–3.06]	1.12 [0.29–4.41]	-	-	-
Hospitalization for TB	0.90 [0.36–2.24]	1.20 [0.39–3.71]	0.56 [0.11–2.63]	-	-	-
TB retreatment	2.56 [0.95–7.45]	2.14 [0.58–7.86]	4.17 [0.74–23.4]	3.04 [1.04–5.09] *	1.93 [0.50–7.54]	8.71 [1.00–77.2]
Ignore treatment outcome	2.05 [1.02–4.10] *	2.46 [1.04–5.83] *	1.43 [0.44–4.66]	2.24 [1.09–4.58] *	2.44 [1.01–5.88] *	2.06 [0.57–7.54]
No DOT	1.15 [0.46–2.88]	2.01 [0.75–5.41]	NE	-	-	-
TB treatment > 6 months	0.73 [0.22–2.50]	0.65 [0.15–2.91]	0.99 [0.12–8.41]	-	-	-

HAP: household air pollution; TB: tuberculosis, DOT: directly observed treatment, Ref = reference; OR: odds ratio, significant ORs in bold, NE not estimable; **^§^**Adjusted for (when not studied) age, sex, source of domestic energy, cooking time, length of TB treatment, retreatment for TB, awareness of treatment outcome; significant ORs in bold; empty cells (-) indicate variables not included in the final model. *p*-value level of significant: * *p* < 0.05; ** *p* < 0.01; *** *p* < 0.001.

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
