# Peer review of "Household Air Pollution Is Associated with Chronic Cough but Not Hemoptysis after Completion of Pulmonary Tuberculosis Treatment in Adults, Rural Eastern Democratic Republic of Congo"

_ijerph, 2018, doi:10.3390/ijerph15112563_

Reviewer 1 Report

The work is good to present the contribution of health effect due to household air pollutant. But there is still something to argue.

The household air quality should be affected by multiple situations. In this article cooking hours, kitchen ventilation, unclean energy, tobacco smoke, vermin/pests in home might all be reasons. But from table 3 and 4, we could see only one or two of those reasons had higher level of significance related with chronic cough and hemoptysis, with the p<0.05. May the authors have more discussion and explanation.

Author Response

Reviewer 1’s comments: The household air quality should be affected by multiple situations. In this article cooking hours, kitchen ventilation, unclean energy, tobacco smoke, vermin/pests in home might all be reasons. But from table 3 and 4, we could see only one or two of those reasons had higher level of significance related with chronic cough and hemoptysis, with the p<0.05. May the authors have more discussion and explanation. 

Reply: We agree that, in this rural setting, quality of household air should be affected by many factors notably by the above cited. On the one hand, by bringing all these different risk factors for chronic cough in the same logistic model (i.e. after adjusting for known confounding variables), cooking hours and use of unclean energy remained statistically and independently associated with chronic cough. Their independent risk remained even after holding constant other variables related to the history of tuberculosis in the model. On the other hand, although not statistically significant (at p<0.05), in this low tobacco smoking area, tobacco smoke exposure tended to increase the risk of chronic cough by 21% particularly among males. We also note that the lack of ventilation was associated with about 101% (upper limit) of risk of chronic cough, especially among women. The true effect might therefore be diluted by other co-variates in the model. In addition, some kitchens with broken wall, for example, might have been noted as not ventilated due to the lack of chimney or of window. This might also explain the lack of significance between indoor or outdoor kitchen practices. We have added this comment under study limitations and we thank you for pointing out the clinically significant effect of these co-variates.

Correction in the text: “Exposure quantification in the kitchen as well as urinary biomarkers of exposure to tobacco smoke could specifically help overcoming ascertainment bias that might occur, since some rural kitchens might present a broken wall in one side and as people tend to underreport their tobacco smoking habits.”

Reviewer 2 Report

This is a well written paper with interesting findings. 

Minor Editorial Comments would be suggestions to improve the methods section. In an effort to keep the methods concise it appears that authors just listed variables, some effort to write in prose would help this section of the methods. For example, – lines 152-154 – consider revising to improve readability, and again in "2.6 Outcome variables"- consider writing a complete paragraph. 

Results appear to be gender specific and women are more vulnerable compared to men based on increased use of cook-stoves for >3 hrs per day in this group, yet this is not something at all discussed by the authors. I think it is an important element to discuss in the paper that is not considered. 

Second, chronic cough seems to be a logical outcome to consider, but it seems that hemoptysis, based on the outcomes of this study, are really mostly related to multiple bouts of TB treatment, and this outcome seems to be a bit more severe than chronic cough. Authors should spend some time justifying their inclusion of this as a respiratory outcome in the study. And/or provide some additional details to the discussion about this issue. 

Author Response

Reviewer 2’s comments: #1: Minor Editorial Comments would be suggestions to improve the methods section. In an effort to keep the methods concise it appears that authors just listed variables, some effort to write in prose would help this section of the methods. For example, – lines 152-154 – consider revising to improve readability, and again in "2.6 Outcome variables"- consider writing a complete paragraph. 

Reply: We have now revised our manuscript accordingly.

Correction in the text:

Line 152-154: “To score for socioeconomic status (SES), we did not use conventional scores because some low-cost devices are widely available in this setting. E.g.: one might find a radio or a mobile phone of around USD 5 to 10 that do not reflect the true SES of the study participants”.

Section 2.6. Outcome Variables “Respiratory symptoms were defined on the basis of responses from the study participants  interview. Symptoms included chronic cough and hemoptysis. To characterize chronic cough, PTB survivals were asked to report any cough of Âł4 weeks duration since completing  TB treatment. To report for hemoptysis, PTB survival were asked to report if they had seen any blood in the sputum since completing their TB treatment.”

Reviewer 2’s comments: #2: Results appear to be gender specific and women are more vulnerable compared to men based on increased use of cook-stoves for >3 hrs per day in this group, yet this is not something at all discussed by the authors. I think it is an important element to discuss in the paper that is not considered. 

Reply: We have revised the discussion session accordingly

Correction in the text: “Tobacco and biomass smoke are likely to share the same composition as they are both generated from combustion of plants. Mechanistically, as does tobacco smoke, biomass smoke also increases the expression of some of the same matrix metalloproteinases. In addition, as for tobacco, our study underscores the effect of exposure-time (time spent in the kitchen for >3hrs per day) in the occurrence of respiratory symptomatology. The vulnerability of women for chronic exposure to indoor biomass and kerosene smoke has been abundantly demonstrated in LMICs [16,17].  In general, hormonal (menopause vs not) and biological status (higher inspiratory flow in women vs men) and speculative differences in epithelial response might explain this vulnerability observed in women compared to men. Moreover, specific determinant related to LMICs setting that might explain women’s vulnerability to HAP should be highlighted. First, due to socio-economic structure, women in LMICs are more likely to be nearer the source of HAP than do men. They are responsible for cooking and might spend more than seven hours daily by the fire. Second, those exposed to HAP are at higher risk of presenting systemic inflammation than those using liquid petroleum gas. Finally, yet importantly, women in LMICs are at risk of suffering from anaemia (multiple pregnancy) that might increase susceptibility to infection if exposed to HAP.”

Reviewer 2’s comments: #3: Second, chronic cough seems to be a logical outcome to consider, but it seems that hemoptysis, based on the outcomes of this study, are really mostly related to multiple bouts of TB treatment, and this outcome seems to be a bit more severe than chronic cough. Authors should spend some time justifying their inclusion of this as a respiratory outcome in the study. And/or provide some additional details to the discussion about this issue. 

Reply:  We agree that hemoptysis is more severe than chronic cough and should merit a particular attention in the discussion. In this population- and questionnaire-based study, hemoptysis might be a proxy of other major post TB treatment complications (chronic aspergillosis, bronchiectasis etc.). Following this observation, the paragraph on hemoptysis is now revised accordingly.

Correction as in the text:

“In DRC, the burden of PTB is still high in spite of the global effort to reduce PTB [25]. In this study, major factors associated with hemoptysis were related to PTB morbidity. TB can cause chronic impairment of lung function, the severity of which increases with the number of episodes of diseases[12]. This is compatible with the independent association we found between hemoptysis and retreatment. In retreated PTB patients, the shrinking of immune function and the dysbacteriosis resulting from the repetition of using large spectra antibiotics both favor fungal colonization. In environments in which patients are frequently re-infected such as in this rural area, pulmonary impairment progressively worsens with each TB episode[7]. In this study, men ignoring their clinical TB outcome tended to present hemoptysis compared to women and were more likely at increased risk of presenting hemoptysis compared to their counterparts.  Our results are comparable to previous findings showing the high rate of default in men compared to women[26], which might jeopardize their clinical outcome[27] and lead to further complication[28]. Hemoptysis can, therefore, be a proxy of post PTB complication such as aspergillosis/ mycetoma, bronchiectasis etc. In this medically underserved area, we had no diagnostic facilities such as radiology, bronchoscopy, spirometry and laboratory facilities, to characterize the various possible structural anomalies and functional processes underlying chronic cough or hemoptysis. Consequently, no specific treatment can be offered either. Although the background prevalence of chronic respiratory symptoms in the source population is not known, the fact that more than 50% of our adult PTB survivors reported chronic cough and 8% reported hemoptysis does represent an extra threat for the fragile health system. PTB survivors who continue to have respiratory symptoms are often treated again for so-called “bacteriologically negative TB”, whilst their symptoms may simply be due to or exaggerated by their HAP exposure in addition to the post-TB sequelae.”

Reviewer 3 Report

This is an interesting study.  

Major comments

In the discussion the reoccurrence of TB in this setting needs to be described, and perhaps added to the limitations of the study (ie was cough caused by new infection)

If I have understood correctly men have similar exposure to pollution (same cooking hours), but no association with cough?  If this is the case some discussion needs to be included around this.

To help the readers in the introduction the need for indoor heating shoul be described (eg average nighttime temperature is 4oC)

Minor comments  

In the paper too many abberviations are used, mostly only a few times each. To improve readability I would avoid this.  The word “underserved” in the discussion could be changed to a different word that better describes the situation.

Author Response

Reviewer 3’s comments: Major comments #1: In the discussion the reoccurrence of TB in this setting needs to be described, and perhaps added to the limitations of the study (ie was cough caused by new infection).

Reply:  we have revised both the discussion and limitation sessions accordingly. However, in this post-conflict area of the DR. Congo, data regarding TB in general are rare and many findings of our TB team are still not published.

Correction in the text:

Discussion section: . In this setting, data on retreatment are scarce. Preliminary data comparing drug resistant (n=142) to drug susceptible (n=1366) PTB patients have demonstrated a high rate of retreatment after earlier default or failure (21 vs 3%) and of having a history of ≥3 previous episode of TB (16 vs 1%).

Limitation section: Also, future studies should consider excluding a new episode of TB and including pulmonary function testing to improve inference to chronic cough and at least blood spot quantification to confirm hemoptysis.

Reviewer 3’s comments: #2: If I have understood correctly men have similar exposure to pollution (same cooking hours), but no association with cough?  If this is the case some discussion needs to be included around this.

Reply:  Following table one, cooking >3h/day was significantly higher in females compared to males (43 vs 31%, p=0.015).  In the text, we have now put this  in bold.

Reviewer 3’s comments: #3: To help the readers in the introduction the need for indoor heating should be described (eg average night-time temperature is 4oC).

Reply: In this rural and post-conflict region, it can be tricky to find accurate meteorological data. From our experience in the region, people use biomass fuel for all domestic related energy. The region can indeed be cold due to several mountains (Mitumba mountain chain) or due to the rainy season. We have inserted this in the introduction session.

Correction in the text:

“In DRC, with the rural electrification rate of 0.4%, population have relied on biomass fuel for cooking and even for heating during night time (mountain regions) and rainy season.”

Reviewer 3’s comments: Minor comments  #3: In the paper, too many abbreviations are used, mostly only a few times each. To improve readability I would avoid this.  

 Reply: Thanks for pointing out this. Following abbreviations have been removed to improve readability: NTB, QALYs, ALTB and BC

Reviewer 3’s comments: #4:  The word “underserved” in the discussion could be changed to a different word that better describes the situation.

Reply: The group of words containing “underserved”: “medically underserved area” have been changed to: “Poor remote setting”